# Host-mediated selection impacts the diversity of *Plasmodium falciparum* antigens within infections

Angela M. Early[1,2], Marc Lievens[3], Bronwyn L. MacInnis[1,2], Christian F. Ockenhouse[4], Sarah K. Volkman[1,2,5], Samuel Adjei[6], Tsiri Agbenyega[6], Daniel Ansong[6], Stacey Gondi[7], Brian Greenwood[8], Mary Hamel[9], Chris Odero[9], Kephas Otieno[9], Walter Otieno[7], Seth Owusu-Agyei[8,10,11], Kwaku Poku Asante[10], Hermann Sorgho[12], Lucas Tina[7], Halidou Tinto[12], Innocent Valea[12], Dyann F. Wirth[1,2] & Daniel E. Neafsey[1,2]

Host immunity exerts strong selective pressure on pathogens. Population-level genetic analysis can identify signatures of this selection, but these signatures reflect the net selective effect of all hosts and vectors in a population. In contrast, analysis of pathogen diversity within hosts provides information on individual, host-specific selection pressures. Here, we combine these complementary approaches in an analysis of the malaria parasite *Plasmodium falciparum* using haplotype sequences from thousands of natural infections in sub-Saharan Africa. We find that parasite genotypes show preferential clustering within multi-strain infections in young children, and identify individual amino acid positions that may contribute to strain-specific immunity. Our results demonstrate that natural host defenses to *P. falciparum* act in an allele-specific manner to block specific parasite haplotypes from establishing blood-stage infections. This selection partially explains the extreme amino acid diversity of many parasite antigens and suggests that vaccines targeting such proteins should account for allele-specific immunity.

[1] Infectious Disease and Microbiome Program, Broad Institute of MIT and Harvard, Cambridge, MA 02142, USA. [2] Harvard T.H. Chan School of Public Health, Boston, MA 02115, USA. [3] GSK Vaccines, 1330 Rixensart, Belgium. [4] PATH Malaria Vaccine Initiative, Washington, DC 20001, USA. [5] Simmons College, School of Nursing and Health Sciences, Boston, MA 02115, USA. [6] School of Medical Sciences, Kwame Nkrumah University of Science and Technology, KNUST - Kumasi, Ghana. [7] KEMRI–Walter Reed Project, Kombewa 40102, Kenya. [8] London School of Hygiene and Tropical Medicine, London WC1E 7HT, UK. [9] KEMRI/CDC Research and Public Health Collaboration, Kisumu 40100, Kenya. [10] Kintampo Health Research Centre, Kintampo 200, Ghana. [11] University of Health and Allied Science, PMB 31 HoVolta Region, Ghana. [12] Institut de Recherche en Sciences de la Santé, Nanoro, Burkina Faso/Institute of Tropical Medicine, 2000 Antwerp, Belgium. Correspondence and requests for materials should be addressed to A.M.E. (email: early@broadinstitute.org) or to D.E.N. (email: neafsey@hsph.harvard.edu)

Plasmodium falciparum, the most deadly of the human malaria parasites, has been co-evolving with its human host for at least tens of thousands of years[1]. Over this time frame, the interactions of parasite, vector, and host have left signatures of strong selection on both human and P. falciparum genomes[2–4]. These evolutionary marks inform our understanding of how P. falciparum successfully invades multiple host and vector tissues while avoiding immune clearance. In turn, this knowledge can be used to develop novel ways of disrupting the P. falciparum transmission cycle. For instance, the high polymorphism of certain parasite proteins reflects their status as immune targets, suggesting that a vaccine composed of the same proteins could similarly elicit an effective immune response and impede infection.

Whole-genome sequencing on a population scale has used signatures of directional or balancing selection to identify immunogenic P. falciparum genes as potential vaccine candidates[5]. While these studies are useful for initially marking genes of interest, such population genetic analyses provide a relatively coarse-grained view of evolution; they identify the composite signals of diverse intra- and inter-host selection pressures, not selection directly mediated by individual vectors or hosts. To better inform therapeutic efforts, we would ideally decouple these various selection pressures by studying selective pressures imposed during different life-cycle stages in isolation.

Here, we describe a high-resolution approach for studying host interactions with P. falciparum, using high-depth sequencing of PCR amplicons and population genetic approaches that treat each infection as a discrete parasite population. In regions of high parasite prevalence, malaria infections often contain multiple haploid lineages of this sexually reproducing eukaryote. We investigate signatures of within-host selection on these parasite populations by using targeted deep population sequencing of three highly polymorphic genes that are known targets of B-cell or T-cell mediated immunity: CSP, TRAP, and SERA2[6–9]. CSP and TRAP are highly expressed by sporozoites[10], the form of the parasite that is injected into the human host by the mosquito and invades the host liver. In contrast, SERA2 is a member of the serine repeat antigen gene family, and like its homologs, is expressed during the disease-causing blood-stage of the parasite life cycle. Therefore, while all antigenic, these genes likely experience different selection regimes due to their divergent expression patterns across the P. falciparum life cycle (Fig. 1). CSP and TRAP are both vaccine development targets. The utility of SERA2 as a vaccine construct remains unexplored, but its high polymorphism across sub-Saharan Africa suggests that host immune recognition of this protein may affect parasite fitness[11].

In addition, this high polymorphism facilitates analysis of intra-host dynamics by allowing us to differentiate a large number of unique haplotypes.

Several previous studies have examined the population-level diversity of CSP and TRAP[12–21]. These genes exhibit high levels of non-synonymous polymorphism in sub-Saharan Africa, and TRAP, in particular, has generally shown consistent evidence of strong selection (but see Barry et al[18].). Several studies have documented functional evidence that natural and vaccine-mediated human immune responses differentially recognize variant CSP peptides[11,22,23]. Resolving the extent and nature of allele-specific immune recognition of these and other P. falciparum antigens would greatly aid the development of effective next-generation malaria vaccines. To date, however, observations of amino acid diversity have not been readily translated into an increased understanding of immune function. One complicating factor is that CSP and TRAP perform essential functions in both the mosquito vector and human host[24]. Ascribing selective pressures and functional constraint exclusively to the human host calls for an approach that decouples the selection imposed at these two distinct life-cycle stages.

Immune selection on P. falciparum is complex; the parasite has developed numerous immune evasion strategies[25] and natural immune protection against malaria is mediated by recognition of a large number of antigens[26]. Dissecting host immune recognition with an evolutionary approach therefore requires a strategy capable of detecting relatively small selection coefficients through deep sampling of both the host and parasite populations. Multi-lineage P. falciparum infections are commonly characterized using small panels of putatively neutral SNPs[27], microsatellite markers[28], or size-length polymorphisms[29,30]. These approaches can quantify the number of clonal lineages within an infection[31,32], but are of limited use for understanding the mechanics of within-host selection. Whole-genome shotgun sequencing has begun to provide information at the nucleotide level[33–35], but the predominance of host DNA in clinical samples limits the depth of parasite genome sampling. This means that low frequency variants within an infection are likely missed, constraining the use of these datasets to questions involving large selection coefficients.

In this study, we use deep sequencing of PCR amplicons to profile the parasite lineages in over 1600 infections. This approach provides unprecedented power to detect selection events (in the form of blocked infections) with small effect sizes via the capacity to resolve the phasing of variants into haplotypes representing distinct lineages. We show evidence of host-imposed selection occurring at both the sporozoite/liver stage and the blood stage of infection. Additionally, despite the large number of

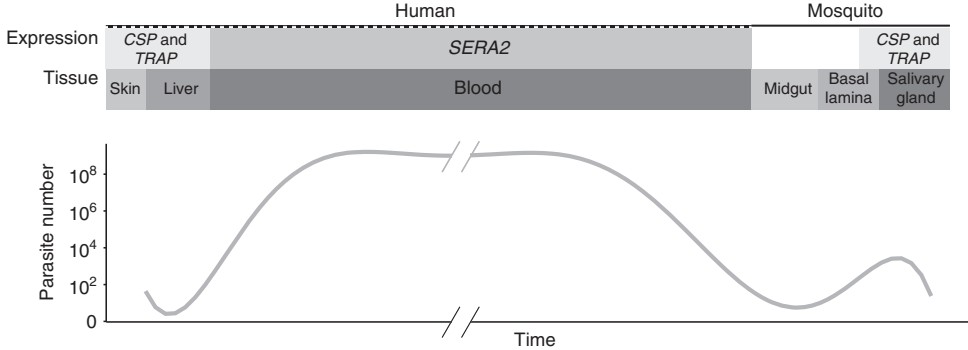

**Fig. 1** Schematic representation of CSP, TRAP, and SERA2 expression profiles. The three genes have contrasting expression patterns that affect the source and duration of the selection pressures they experience. CSP and TRAP are among the most abundantly expressed proteins in the sporozoite stage[10]. SERA2 expression is highest during the blood stage and is in the 50th percentile of genes expressed in schizonts[65]

**Table 1 Population genetic statistics for *CSP*, *TRAP*, and *SERA2* calculated using genome-wide sequencing data**

| | CSP | | TRAP | | SERA2 | |
|---|---|---|---|---|---|---|
| | **Full gene** | **Amplicon region** | **Full gene** | **Amplicon region** | **Full gene** | **Amplicon region** |
| Nucleotide diversity | | | | | | |
| Senegal | 0.00601 (99) | 0.0204 (99) | 0.0102 (99) | 0.0122 (99) | 0.00191 (96) | 0.00888 (99) |
| Malawi | 0.00806 (99) | 0.0211 (99) | 0.0102 (99) | 0.0132 (99) | 0.00198 (96) | 0.00894 (99) |
| Non-synonymous nucleotide diversity | | | | | | |
| Senegal | 0.00674 (99) | 0.0254 (99) | 0.0120 (99) | 0.0137 (99) | 0.00181 (96) | 0.0120 (99) |
| Malawi | 0.00713 (99) | 0.0263 (99) | 0.0120 (99) | 0.0144 (99) | 0.00197 (97) | 0.0119 (99) |
| Tajima's $D$ | | | | | | |
| Senegal | −1.39 (78) | -0.132 (96) | 0.536 (98) | 0.298 (98) | −1.21 (84) | −0.494 (94) |
| Malawi | −0.765 (94) | 0.519 (99) | 0.712 (99) | 0.257 (99) | −1.24 (89) | −0.868 (94) |
| $F_{ST}$ | | | | | | |
| Senegal–Malawi | 0.0512 (89) | 0.0163 (57) | 0.0576 (91) | 0.0322 (80) | 0.0236 (71) | 0.00716 (22) |

Genome-wide percentiles are given in parentheses

antigens that drive acquired immunity to malaria, we are able to point to specific amino acid positions that show a high probability of mediating allele-specific host recognition within two of these individual antigens, suggesting that the impact of acquired immunity on parasite fitness and infection outcome can be dissected with high-resolution genomic data.

## Results

**High population-level diversity suggests immune selection.** Genes that experience immune selection are likely to exhibit high levels of non-synonymous polymorphism and site frequency spectra that indicate the action of balancing or diversifying selection. Both of these expectations hold for our three focal genes: *CSP*, *TRAP*, and *SERA2* (Table 1). To obtain a picture of these three genes in the context of genome-wide diversity patterns, we analyzed whole-genome *P. falciparum* sequence data from single-lineage infections in Senegal ($n = 99$) and Malawi ($n = 110$). All three of the focal genes are among the 5% most polymorphic genes in both populations, with *CSP* and *TRAP* being among the top 1%. The regions selected for amplicon sequencing display even higher levels of polymorphism than the full genes (Fig. 2a). Relative to other coding regions in the *P. falciparum* genome, the site frequency spectra of these genes and amplicon regions are skewed toward having a high proportion of intermediate-frequency SNPs, as measured by Tajima's $D$, indicative of diversifying selection. *TRAP*, as well as the regions of the *TRAP* and *CSP* genes chosen for amplicon sequencing, are within the upper 95th percentile of both populations' Tajima's $D$ distributions (Fig. 2b). The genes, however, do not show evidence of extensive adaptation to specific geographic locations or host populations. None of the genes showed notable inter-population divergence as measured using $F_{ST}$, and each amplicon region showed lower population differentiation than its corresponding full gene (Fig. 2c).

PCR-based amplicon sequencing was conducted on infection samples collected across five study sites in the context of a clinical vaccine trial (RTS,S/AS01)[11]. Each amplicon was composed of ~300 nt of contiguous sequence, acquired from overlapping Illumina MiSeq read pairs. On average, thousands of reads were sequenced per amplicon per sample, providing a more fine-grained view of nucleotide diversity within the three focal genes. These data reveal a number of low frequency variants that were not detected in previous sequencing studies, but the overall patterns of polymorphism across the five study sites were consistent with what was observed in the Senegal and Malawi data (Table 2). The *CSP* amplicon contains three regions of high

amino acid diversity[11] that include two previously described T-cell epitopes (Th2R and Th3R)[8] and a B-cell epitope (DV10)[7]. These epitopes physically co-localize to one side of the folded protein and surround a conserved hydrophobic pocket with extremely low amino acid diversity[36,37]. For the *TRAP* and *SERA2* amplicons there are no available protein structures, but their patterns of diversity are similar to *CSP*, showing discrete stretches of high diversity linked by regions with few or no variants (Fig. 3).

As observed in the genome-wide data, population differentiation ($F_{ST}$) between the western and eastern African study sites is not unusually high for any of these amplicons, although there are a few individual variant sites that show reasonably high population differentiation (Supplementary Data 1−3). Interestingly, the two most diverse nucleotide sites in *CSP* (nt965 and nt952) show little population differentiation. These positions respectively contain four and three segregating nucleotides, and both have a pairwise nucleotide diversity ($\pi$) above 0.6 in all five study sites. Still, all population pairs show an $F_{ST}$ under 0.01 at both positions suggesting they experience similar selection pressures at all geographic locations.

Because it provides haplotypic information, the amplicon sequencing approach also allows for an analysis of linkage disequilibrium (LD), or the preferential association of specific allelic variants on the same genetic background. We found that LD is highest for *CSP* and lowest for *SERA2* (Fig. 4; Supplementary Fig. 1). While previously noted in the Th2R and Th3R regions, we found that this LD extends to the DV10 region as well. Interestingly, LD in *CSP* exists both within and between known antigenic regions. Compared to *TRAP* and *SERA2*, this level of LD is high given the genetic distance between these regions (Supplementary Fig. 2) and cannot be attributed to recent mutations and the expansion of a single haplotypic background (Supplementary Data 4). Previous work has found no evidence that intra-molecular forces drive the skewed association of alleles at specific amino acid positions on *CSP* haplotypes[13]. This suggests that the maintenance of preferred allelic combinations at these positions might reflect the action of protein–protein interactions, which could include antibody recognition of conformational epitopes. The lower levels of LD found in *TRAP* and *SERA2* suggests that these proteins may experience fewer structural constraints or be subject to immune recognition through linear rather than conformational epitopes.

**Intra-infection diversity reflects host selection.** The majority of infections sequenced with the amplicon approach contained multiple haplotypes (*CSP*: 60.0%; *SERA2*: 53.6%; *TRAP*: 53.0%;

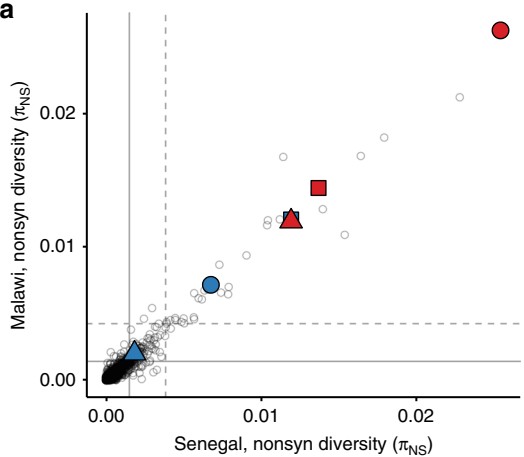

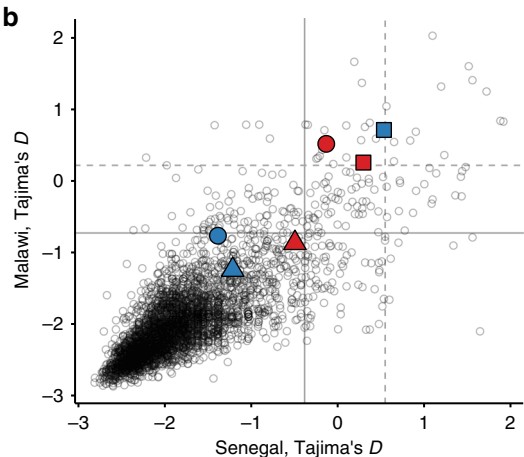

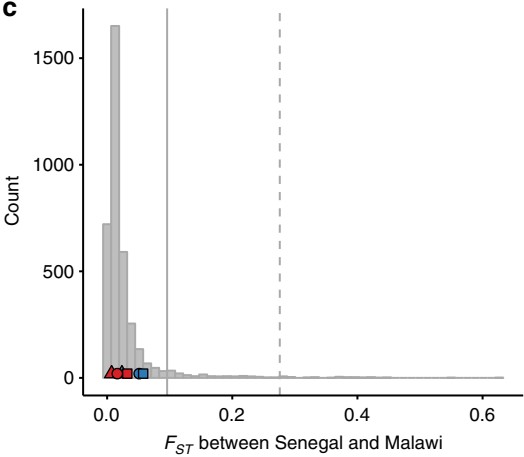

**Fig. 2** Genome-wide signatures of *P. falciparum* polymorphism in Senegal and Malawi. **a** *CSP*, *TRAP*, and *SERA2* show unusually high levels of nonsynonymous nucleotide diversity. **b** *TRAP*, the *TRAP* amplicon region, and the *CSP* amplicon region have unusually high Tajima's *D* values. **c** None of the genes or amplicon regions show strong population differentiation ($F_{ST}$). Each point represents the coding sequence from a single protein-coding gene. Solid and dashed lines mark the 95th and 99th percentiles, respectively. Focal genes are marked with filled symbols: *CSP* (circle), *TRAP* (square), *SERA2* (triangle). Full genes are colored blue, whereas the regions selected for amplicon sequencing are colored red

Supplementary Fig. 3). We wished to determine whether these polyclonal infections were composed of haplotypes drawn at random from the population, or if selection was acting to group similar (or dissimilar) haplotypes together. Such selection could have occurred within the observed infection, or, as Hastings[38] demonstrated, could represent the cumulative effect of past selection events across multiple hosts as haplotype associations can be maintained through stable co-transmissions between individuals. To test for the presence of selection, we simulated a set of 10,000 replicate datasets via bootstrapping of the pooled haplotypes from each study site (Fig. 5). We could not determine whether natural infections contained multiple parasite lineages with identical haplotypes, so we ensured that the simulated infections did not contain duplicate haplotypes. We then calculated the within-infection diversity of each infection and asked whether haplotypes within observed infections were more similar to one another than haplotypes in simulated infections. Our metric of diversity is based on a population genetics framework and depends on the number of distinct alleles at each variant amino acid position within each infection. As shown in Fig. 5, the number of alleles at a given amino acid position can differ from the infection haplotype count since distinct haplotypes may share the same allele.

We found that observed within-infection amino acid diversity was lower than expected for all three genes across the five study sites (Student's *t*-test with Fisher's combined *P*, $P = 0.0183$ (*CSP*), $P = 0.0123$ (*TRAP*), and $P = 0.0219$ (*SERA2*)). Selection, however, does not need to be evoked to explain this pattern; a reduction in observed heterozygosity is also expected under a model of population sub-division (the Wahlund effect). While we earlier observed that population differentiation is relatively low between study sites (Fig. 3), *P. falciparum* populations are known to contain microstructure at the level of a household or village[39], which could contribute to protein-wide reductions in within-infection diversity. However, while population sub-structure would cause an overall decrease in heterozygosity across the given protein, the change in heterozygosity at any individual site would fluctuate more widely. We, therefore, looked for amino acid variants that showed decreased within-infection heterozygosity across all five geographic sites. This pattern would suggest the consistent action of selection rather than the random effects of genetic drift operating on structured sub-populations.

Across all three amplicons, five amino acid positions showed statistically significant reductions in diversity (Fig. 6). Interestingly, the amino acid position identified in *CSP* (aa354) is one of the seven amino acid positions that showed significant differential protective efficacy in a phase 3 trial of the malaria vaccine RTS,S/AS01[11]. In Neafsey et al.,[11] this was observed as a vaccine-induced effect that manifested as greater protection against infection by parasites harboring a *CSP* haplotype matching the vaccine construct in RTS,S/AS01-vaccinated individuals. In this study, we limit our analysis of *CSP* to individuals who were enrolled in the control arm of the trial and therefore only detect the effects of natural immunity. Combined, these studies provide two independent lines of evidence suggesting that this amino acid position is recognized in an allele-specific manner by an adaptive B- and/or T-cell response.

Given past evidence that specific haplotypic combinations enhance parasite infectivity[40], we also considered the inverse question: do certain genotypes preferentially associate within infections? Only one amino acid position (*CSP* aa301) showed evidence of increased within-infection heterozygosity. As with *CSP* aa354—which showed increased within-host homozygosity—*CSP* aa301 was identified by Neafsey et al[11]. as showing an effect of allele-specific vaccine efficacy in the RTS,S/AS01 vaccine trial. One of the nucleotide positions coding for this amino acid

**Table 2 Summary of nucleotide variants described with targeted amplicon sequencing**

| Amplicon region (amplicon length) | Sampled infections | Sequenced amplicons | Total variant sites | Nonsyn. variant sites | Singleton variants | Common variants[a] | Highly variant sites[b] |
|---|---|---|---|---|---|---|---|
| *CSP* (288 nt) | | | | | | | |
| Total | 1687 | 3821 | 55 | 50 | 15 | 23 | 12 |
| West Africa | 935 | 2147 | 46 | 43 | 8 | 24 | 8 |
| East Africa | 752 | 1674 | 41 | 39 | 8 | 23 | 8 |
| *TRAP* (318 nt) | | | | | | | |
| Total | 4209 | 8774 | 52 | 52 | 13 | 15 | 5 |
| West Africa | 2334 | 4925 | 38 | 38 | 7 | 16 | 5 |
| East Africa | 1875 | 3849 | 38 | 38 | 10 | 15 | 3 |
| *SERA2* (258 nt) | | | | | | | |
| Total | 4209 | 9007 | 67 | 67 | 19 | 12 | 16 |
| West Africa | 2334 | 5048 | 53 | 53 | 17 | 13 | 12 |
| East Africa | 1875 | 3959 | 49 | 49 | 11 | 8 | 11 |

[a] Minor allele frequency >0.02
[b] >2 segregating alleles

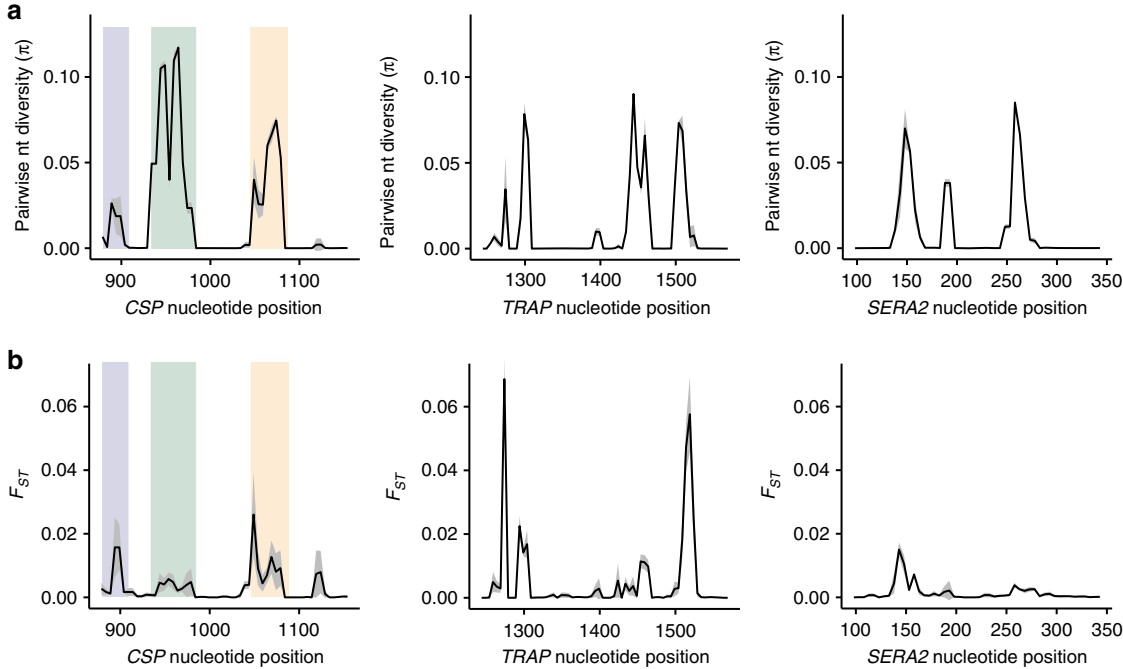

**Fig. 3** Nucleotide diversity and population differentiation within *CSP*, *TRAP*, and *SERA2* as measured with amplicon sequencing. **a** Graphs of pairwise nucleotide diversity ($\pi$) present the mean ± 1 S.D. across all five study sites. **b** Graphs of population differentiation show the mean ± 1 S.D. of pairwise $F_{ST}$ values between western and eastern study sites. Statistics were calculated using SNPs within overlapping 10-nt windows with a step size of 5 nt. The colored bars in the *CSP* graph mark three previously identified regions of high diversity: DV10 (blue), Th2R (green), and Th3R (yellow)

(nt902) is also the *CSP* nucleotide position with the highest divergence ($F_{ST}$) between western and eastern African study sites. Overall, however, the relatively low number of sites showing increased heterozygosity suggests that immune interference does not widely contribute to changes in infection diversity—at least not at the level of individual amino acid variants. Instead, the stronger evolutionary force in these populations appears to be allele-specific selection against particular genotypes, leading to an overall decrease in within-infection diversity.

**SERA2 amino acid diversity is lower in older patients**. The above analysis suggests that either the host or mosquito vector causes infections to be selectively assembled. To further test the

hypothesis that selection is specifically exerted by adaptive immune responses, we investigated how infection diversity relates to a known correlate of immune competence: patient age. In malaria-endemic countries, adults do not acquire full immunity to malaria infection[41], but the parasite levels in their blood are significantly reduced compared to the levels in young children, and they are less likely to develop clinical malaria[42]. The acquisition of adaptive immune defenses plays a role in this age-dependent disease resistance[43], although the exact mechanism remains unclear. Our dataset offers an opportunity to examine the effects of age-related immune acquisition on a nucleotide level. To do so, we make two assumptions: (1) hosts mount more effective immune responses against parasite haplotypes to which they were previously exposed; and (2) patient age positively

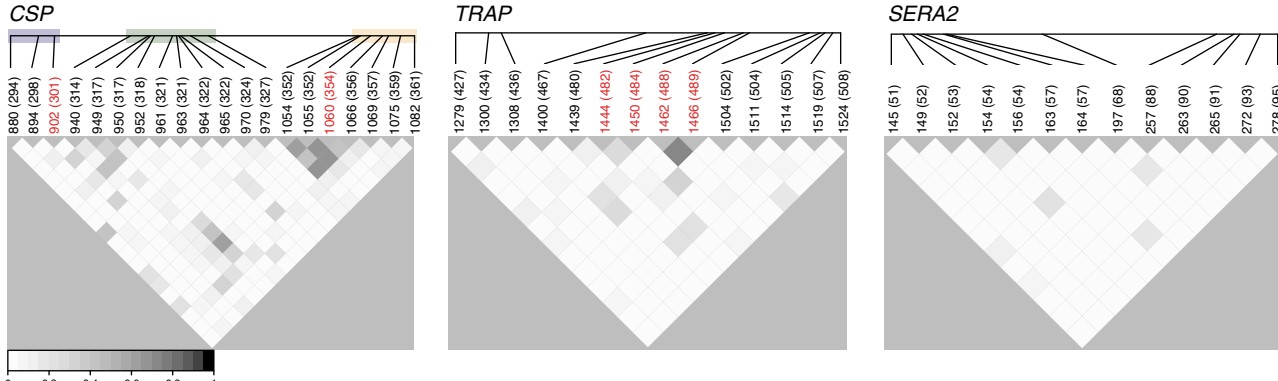

**Fig. 4** Linkage disequilibrium (LD) within the *CSP*, *TRAP*, and *SERA2* amplicon regions. LD between polymorphic nucleotides was calculated as Q*, an extension of $r^2$ that allows for multiple alleles per site. Corresponding amino acid positions are in parentheses. Positions marked in red showed evidence for altered intra-host diversity at the amino acid level. For *CSP*, shading marks the positions within the DV10 (blue), Th2R (green), and Th3R (yellow) epitope regions. Only nucleotide positions with a major allele frequency <0.98 were included in the analysis. Data shown are from Nanoro, Burkina Faso. Observed LD trends were consistent across all five study sites (Supplementary Fig. 1)

correlates with cumulative exposure to *P. falciparum*. Under this model, successful infection blocking would be haplotype-specific and more frequent in older relative to younger children. This would lead to lower infection diversity in the former relative to the latter group at amino acid positions that are effective immune targets.

To test if patient age correlates with *P. falciparum* infection diversity, we constructed a quasi-Poisson model that accounted for additional factors including study site, complexity of infection, and type of sampling (active vs. passive). For SERA2, but not CSP or TRAP, we found a negative effect of patient age on within-infection amino acid diversity (−0.032 mismatches per year, quasi-Poisson model, $P = 0.028$; Table 3). Across the SERA2 amplicon, mean number of pairwise differences per infection was 2.907. This suggests that as children age by one year, the amino acid diversity at SERA2 reduces by ~1.1%.

We next modeled diversity at individual amino acid positions within each amplicon. Diversity at two amino acid positions in SERA2 (aa93 and aa95) showed nominally significant age effects (aa93: −0.12 mismatches per year, quasi-Poisson model, $P = 0.047$; aa95: −0.39 mismatches per year, quasi-Poisson model, $P = 0.032$), but their statistical significance did not survive multiple-testing correction. Overall, therefore, this result suggests that the reduction in amplicon-level diversity cannot be attributed to the effect of a single amino acid site. This is consistent with a model where immune recognition at this antigen occurs at multiple sites rather than at a single amino acid position. Future studies incorporating longitudinal sampling of individual hosts could resolve whether different hosts preferentially recognize different amino acid positions within this antigen.

Given that *SERA2* is expressed during the protracted blood-stage, this observation is consistent with evidence that natural acquired immunity most strongly controls *P. falciparum* transmission at the blood-stage of infection[44]. Detection of this trend in young children (<5 years of age) suggests that acquired immune defenses likely begin building at an earlier age than previously observed in clinical studies[44]. These defenses, however, may not yet be sufficient to cause measurable reductions in disease incidence.

## Discussion

Targeted deep sequencing of pathogen genes provides a powerful and efficient way to gain insight into host–pathogen interactions. The amino acid positions we identify with this approach may

mediate allele-specific immune responses to *P. falciparum* antigens, including the leading vaccine component CSP. While we do not know the extent to which parasite antigens are or are not cross reactive, several recent vaccine candidates—all monovalent subunit vaccines developed from single *P. falciparum* genotypes— have displayed allele-specific protection[11,45]. In light of these observations, one proposed course of action is the development of multivalent vaccines that target multiple pathogen genotypes[46]. By identifying specific amino acid positions that likely contribute to differential recognition, we provide key information for the development of such next-generation vaccines. Further, we illustrate that vaccine development challenges will be antigen-specific: due to a combination of protein function and expression pattern (Fig. 1), each protein experiences distinct selection pressures and structural constraints that create unique signatures of diversity, divergence, and LD.

While previous genetic analyses have provided evidence that *CSP* is under selection in the human host[12–16], few have provided actionable information on specific amino acid positions that may mediate natural host–pathogen interactions. One important exception is Gilbert et al[40]., who studied within-host haplotype diversity of the CSP T-cell epitope Th3R. Like this study and Neafsey et al[11]., they found that alleles at CSP amino acid 354 show preferential associations within infections. Interestingly, however, their samples showed the opposite effect of the one observed here. In our study, diversity at aa354 is lower within hosts than would be expected. Conversely, Gilbert et al[40]. found that two 8-aa haplotypes (cp26 and cp29) that differed only at aa354 showed the propensity to co-occur within infections, thereby increasing diversity at this single amino acid position. Additional in vitro studies found that these two haplotypes acted in concert to evade immune detection through a process of immune interference, thereby increasing their combined fitness[47]. In our samples, one of these haplotypes was at too low of a frequency to assess its potential preferential associations. Thus, the diversity changes we observed at aa354 were independent of the 8-aa haplotypes described by Gilbert et al[40]. The opposing directions of the effects noted across studies suggests that non-additive interactions between alleles can determine the course of selection on any individual site, reinforcing the need to consider extended haplotype information and patterns of LD when studying immune recognition of *P. falciparum* antigens.

As we demonstrate here, population-level and host-level approaches provide complementary information that together identify not only genes, but also specific amino acid positions of

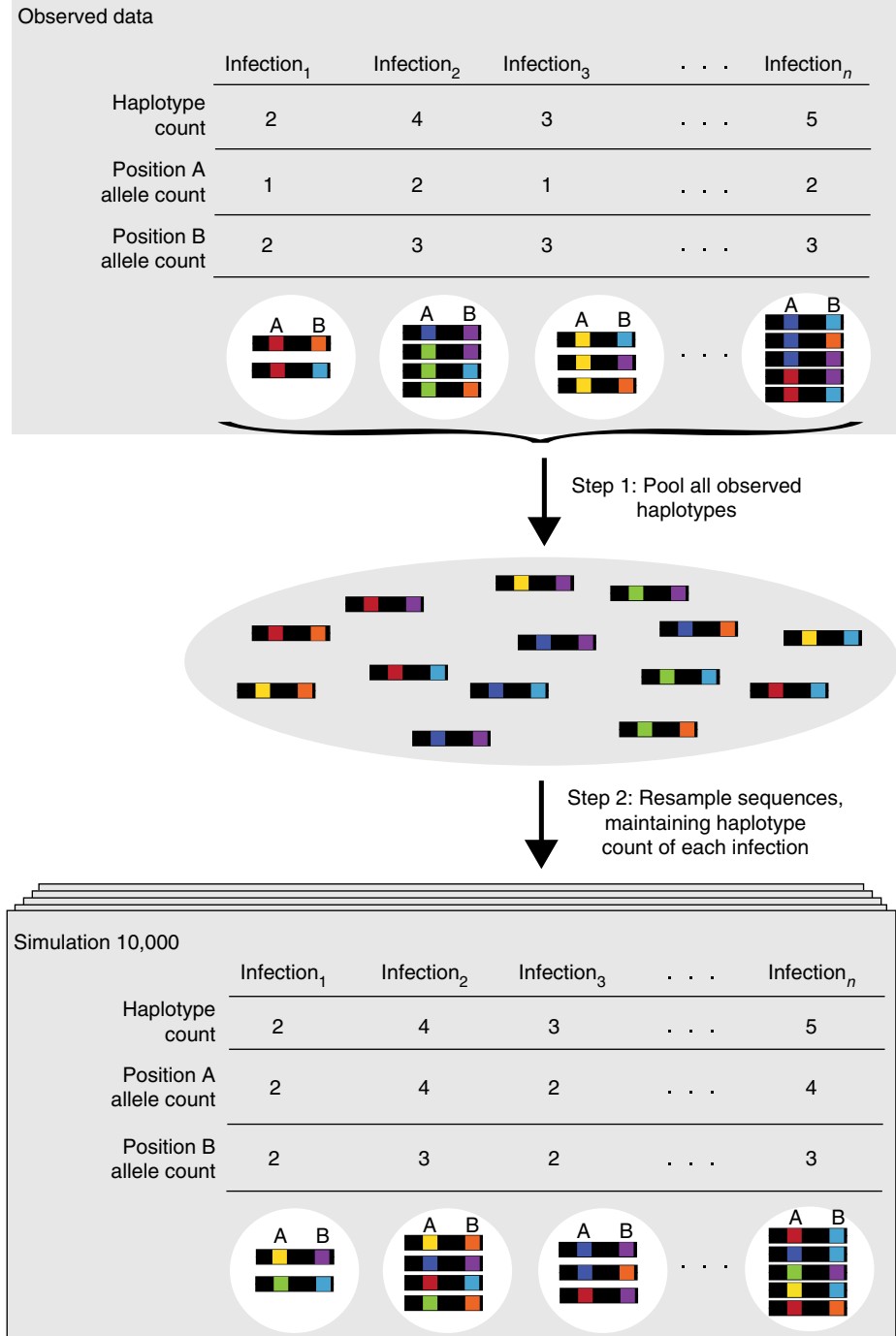

**Fig. 5** Infection simulations. For each study site, we created 10,000 simulated datasets to test whether haplotypes are distributed across infections according to simple random sampling from the population. Haplotype counts within the $n$ infections were held constant while the identity of each haplotype was resampled from the complete pool of sequenced haplotypes. To determine whether selection affects infection composition, we compared the diversity within the observed and simulated infections at each amino acid position and across the entire amplicon

immunological and evolutionary interest. Gene-level statistics like Tajima's $D$ can mask relevant intra-gene patterns, especially when genes include constrained functional elements. Similarly, population-level analyses average selection pressures across multiple life stages and multiple human hosts. By focusing on a single life stage, we find evidence for there being amino acid positions under selection within *CSP* and *SERA2*—genes that do not show significant evidence for selection in a traditional population-level genome-wide analysis. In this regard, there are parallels between the analysis we present here and approaches,

such as selection components analysis[48] and single-cell sequencing. In essence, these are all ways of dissecting a single composite signal to better understand the distinct forces or dynamics underneath. This added granularity helps assign causality and identify targets of selection that may be masked by averaging signals across loci, hosts, or life stages.

The utility of targeted amplicon sequencing therefore extends beyond questions of immune selection. There is growing recognition that intra-host dynamics must be considered when studying diverse aspects of *Plasmodium* biology, including drug

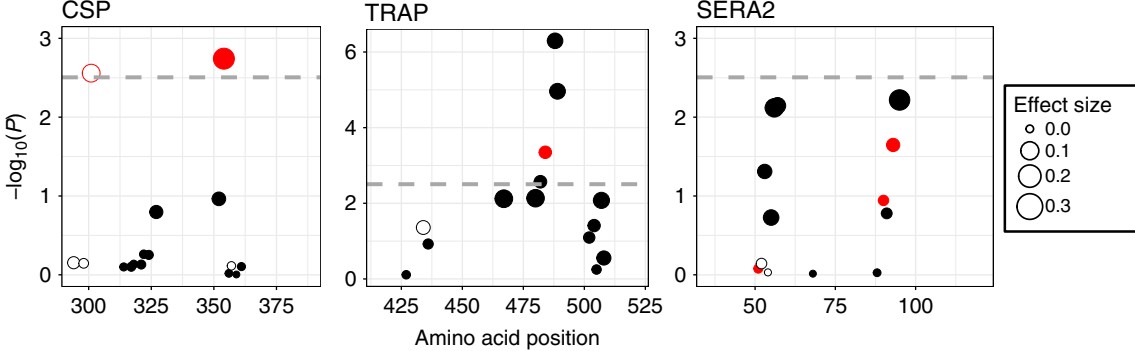

**Fig. 6** Deviation from expected within-host diversity at individual amino acid positions within the *CSP*, *TRAP*, and *SERA2* amplicon regions. Points above the dotted line mark amino acid positions with significant heterozygosity differences between observed and simulated infections after Bonferroni correction. Filled circles show a reduction in observed within-infection diversity whereas open circles show an increase in observed within-infection diversity. The size of the point corresponds to the effect size estimated with a quasi-Poisson regression. Points in red show a marginally significant effect of age on diversity ($P < 0.05$) before multiple-testing correction, but none of these interactions remain significant after Bonferroni correction. Only variant positions with a major allele frequency <0.98 were included in the analysis

### Table 3 Parameter estimates from quasi-Poisson regression model of within-host infection diversity

| Variable | CSP | | TRAP | | SERA2 | |
|---|---|---|---|---|---|---|
| | Effect estimate | P | Effect estimate | P | Effect estimate | P |
| Patient age (in years) | 0.0094 | n.s. | 0.015 | n.s. | −0.032 | 0.028 |
| Sample type[a] | −0.033 | n.s. | 0.016 | n.s. | 0.072 | 0.0015 |
| Haplotype number | 0.011 | n.s. | -0.012 | 0.035 | -0.0020 | n.s. |

[a] Longitudinal, asymptomatic sampling vs. clinical infection sample

resistance[49,50], disease severity[51–54], and transmission rate[55,56]. Our observations show that human-mediated selection on *P. falciparum* is sufficiently strong to be detected using measures of within-infection diversity and demonstrate the utility of incorporating evolutionary approaches into *Plasmodium* studies. Targeted deep sequencing of genes involved in additional biological processes can further augment current efforts to understand and manipulate the interactions between humans and this complex parasite.

## Methods

**Filtering and analysis of genome-wide sequence data**. To place our amplicon sequences within a genome-wide context, we obtained genome-wide variant calls for *P. falciparum* samples collected in Senegal[57] ($n = 137$) and Malawi[19] ($n = 269$) from the Pf3k project (release 5; www.malariagen.net/projects/pf3k). Variant calls were downloaded as vcf files, and only variable sites that were flagged as passing all GATK filters were retained. We further limited the analysis to variant sites that fell within coding regions and used SnpEff calls to annotate each variant as synonymous or non-synonymous. To accurately calculate population-level allele frequencies, we identified samples that likely contained only single *P. falciparum* clones. *Plasmodium* is haploid throughout its human-infecting life stages. Heterozygous calls therefore signal either alignment errors or the presence of multiple co-occurring genetic lineages within a sample. For each sample, we calculated the fraction of variant sites across all genes that were heterozygous or missing. Samples with either ≥2% heterozygous calls or ≥4% missing calls were removed from the analysis. To prevent this calculation from being dominated by poorly aligned genes, we first masked a set of 243 genes where 20% or more samples had either ≥25% heterozygous calls or ≥25% missing calls. This left us with a set of 99 Senegal and 110 Malawi samples that likely contained a single *P. falciparum* genotype (Supplementary Data 5). All downstream analysis was limited to this set of 209 single-clone infections. After identifying these single-clone samples, we returned the previously masked genes to the dataset and re-calculated population-level diversity for each gene. We found 315 genes that had either ≥20% heterozygous calls or ≥20% missing calls in these samples. As these were single-clone samples, high levels of intra-sample heterozygosity or missing data likely signaled alignment errors, not true intra-host variants. We therefore masked these 315 genes from the analysis.

We transformed any remaining heterozygous calls into homozygous calls by retaining only the allele with greatest read support. Only SNP calls were retained for use in downstream analyses.

For each gene, we calculated two measures of nucleotide diversity ($\pi$ and Watterson's $\theta$), Tajima's $D$, and Weir and Cockerham's estimate of $F_{ST}$[58] using custom Perl scripts (available on request). Sites with more than two alleles were retained and counted as multiple segregating sites when calculating Watterson's $\theta$. The numerator and denominator of the $F_{ST}$ statistic were averaged individually across sites in a given gene. At each site, we adjusted the sample size based on the number of samples with successful calls. We assumed that all sites without called variants were reference alleles for all the samples. For each gene, statistics were calculated across all sites, at synonymous sites only, and at non-synonymous sites only. To ensure we were not biasing our results by using genes with little coverage, we limited our analysis to 3682 genes that had at least five SNPs in one of the populations.

**Filtering and analysis of targeted amplicon sequences**. The amplicon sequences for *CSP* (PF3D7_0304600), *TRAP* (PF3D7_1335900), and *SERA2* (PF3D7_0207900) were previously generated and filtered by Neafsey, et al[11]. The *CSP* amplicon covers the diverse C-terminal region of the *CSP* gene, which forms a large portion of the RTS,S vaccine construct (Pf3D7_03_v3:221352.221639)[59]. The *TRAP* and *SERA2* amplicons were designed to capture regions of high diversity within these respective genes (Pf3D7_13_v3:1465059.1465376; Pf3D7_02_v3:320763.321020). Parasite DNA was obtained from blood samples acquired as part of the RTS,S/AS01 vaccine phase 3 trial[60]. In brief, children (5–7 months of age) and infants (6–12 weeks age) were randomly assigned to one of two arms of the vaccine trial. Approximately two-thirds received the RTS,S/AS01 vaccine regimen whereas one-third received a control rabies vaccine. Blood samples included both passive and active sampling. Passive clinical samples were collected from febrile children with at least 5000 parasites/μl at the time of first clinical malaria infection occurring within the first 12 months following completion of the triple-dose vaccination regimen. Active cross-sectional sampling occurred in a subset of participants at 18 months post-vaccination. Samples were stored as dried blood spots until DNA extraction. Targeted regions were then amplified and sequenced using overlapping 250-bp paired-end reads with Illumina MiSeq technology. Paired-end reads were merged using FLASH[61] and aligned to the PlasmoDB v.9.0 3D7 *P. falciparum* reference genome using BWA v.0.74[62].

This targeted amplicon approach allowed the identification of complete, linked haplotypes across the sequenced regions. To minimize the misidentification of spurious haplotypes within samples, Neafsey, et al[11] adopted a set of conservative filters that removed the following: (1) all reads with uncalled bases; (2) sequences representing fewer than 1% of the reads in a given sample; (3) haplotypes observed only once in the entire dataset that were represented by fewer than 500 reads. After filtering, haplotypes within each sample were clustered. If two haplotypes differed by only 1 nucleotide (nt) and their read ratio was less than 0.15, the haplotypes were merged. All samples with more than two haplotypes following clustering and filtering were evaluated for PCR chimerism. If a haplotype could be expressed as a simple combination of two other haplotypes called for the same sample and was supported by fewer than 250 reads, it was eliminated from the final dataset. After merging and filtering, the median read count per sample was still very high (CSP: 5559, TRAP: 3001, SERA2: 4912).

For the purposes of this study, we used data from the five study locations with the highest transmission intensities: Kintampo, Ghana; Agogo, Ghana; Nanoro, Burkina Faso; Kombewa, Kenya; and Siaya, Kenya. In total, this provided information on 4209 infections: 2522 from RTS,S/AS01-vaccinated and 1687 from control-vaccinated individuals. Analyses of CSP used only data from the control-vaccinated individuals. After confirming that the RTS,S/AS01 vaccine had no effect on diversity at the TRAP and SERA2 amplicons, we included all patient samples in analyses of these genetic regions.

**Population genetic and statistical analyses**. For each study site, we calculated pairwise nucleotide diversity (π) and Weir and Cockerham's measure of population differentiation ($F_{ST}$;[58]) at each SNP and within 10 nt windows across each amplicon region. We calculated LD between pairs of sites—including sites with more than two variants—using Lewontin's $D'$[63] and Hedrick and Thomson's $Q^*$[64].

To test for the effect of age on within-infection amino acid diversity, we constructed a quasi-Poisson regression model with patient age at time of infection, study site, sample type (passive/clinical, active/cross-sectional), and haplotype number as fixed effects. Within-infection diversity was modeled separately for each amplicon. Models were constructed to study infection diversity on the level of the whole amplicon and on the level of individual amino acid positions. The analysis was conducted in R with the glm function.

**Modeling expected infection heterozygosity**. Within our sample of 4209 infections, we recovered 0–13 haplotypes per amplicon per infection, with over 50% of infections containing multiple segregating P. falciparum genotypes. Infection diversity is dependent on COI, so we determined the expected level of infection diversity through bootstrapping. For each study site, we created a haplotype pool that contained the exact number of each haplotype recovered during sequencing. We then created 10,000 sets of simulated infections that maintained the same COI structure observed in each population but where the infecting strains were sampled with replacement from the haplotype pool (Fig. 5). As we were not able to differentiate identical haplotypes in the natural infections, each simulated infection was similarly constrained to contain only unique haplotypes.

For each sample (both observed and simulated), we calculated within-infection amino acid diversity for the full length of each amplicon as the mean number of differences between each pairwise haplotype comparison within the sample. When analyzing diversity at individual positions, we counted the number of haplotype pairs that mismatched at the given site. To determine whether mismatches were more or less frequent in the observed vs. the simulated infections at a given amino acid position, we constructed a quasi-Poisson regression model that included dataset (observed, simulated), haplotype number, study site, sample type (passive/clinical, active/cross-sectional), and age group (infant, child) as fixed effects. We additionally included two interaction terms: (dataset × sample type) and (dataset × age group). These accounted for the possibility of increased acquired immunity in older children and for potential differences between symptomatic and asymptomatic infections. The Bonferroni correction was performed to account for multiple testing across multiple sites within each individual gene. Variant sites were only analyzed if they were polymorphic in all five populations and had a global major allele frequency under 0.98. For the simulated data, we included results from 1000 bootstrapped datasets. The analysis was conducted in R with the glm function.

**Data availability**. All raw Illumina reads from the original Neafsey et al[11] study are available on the NCBI Short Read Archive (BioProject PRJNA235895 [https://www.ncbi.nlm.nih.gov/bioproject/?term = PRJNA235895]). Supplementary Data 6 and 7 contain the final called haplotypes and metadata from children and infants enrolled in the control arm of the vaccine trial.

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

## Acknowledgements
We thank the participants of the RTS,S/AS01 phase 3 clinical trial and their parents as well as the members of the RTS,S Clinical Trials Partnership. Aimee Taylor, Seth Redmond, and members of the Neafsey and Wirth research groups provided thoughtful comments and discussions. This project has been funded in whole or in part with Federal funds from the National Institute of Allergy and Infectious Diseases, National Institutes of Health, Department of Health and Human Services, under Grant Number U19AI110818 to the Broad Institute, and in part by a grant from the Bill and Melinda Gates Foundation to D.F.W.

## Author contributions
A.M.E. and D.E.N. conceived of the analysis. A.M.E. conducted the analysis and wrote the manuscript. D.E.N. and D.F.W. provided funding and project guidance. B.L.M. and S.K.V. provided project administration and oversight. M.L., C.F.O., S.K.V., D.F.W., and D.E.N. contributed to the development of original clinical study. S.A., T.A., D.A., S.G., B.G., M.H., C.O., K.O., W.O., S.O., K.P.A., H.S., L.T., H.T., and I.V. executed the original clinical trial and collected the samples. All authors reviewed the manuscript and gave final approval for publication.

## Additional information

**Competing interests:** M.L. is an employee of GSK group of companies, which is involved in malaria vaccine development. The remaining authors declare no competing interests.

