## [Peer Review File · Nature Communications]

Reviewers' comments:

Reviewer #1 (Remarks to the Author):

The manuscript by Early et al. describes the analyses of >2,800 *P. falciparum* infections using targeted amplicon sequencing for three antigens. The amount of data generated is very impressive and the analyses interesting (though they could sometimes be explained more simply and convincingly). I have some specific concerns but, overall, a very exciting study.

I do not understand the emphasis on "balancing selection". For example, on line 146, "diversifying selection" should probably be a better suited terminology than balancing selection, which is very specific and may not reflect the situation described by the authors. Also, balancing selection is classically associated with positive Tajima's D values and two of the three genes targeted have clear negative values (though slightly less negative than the genome average). I would recommend replacing this term throughout the manuscript. In general, I think the paper would be improved by simplifying the discussion on the evolutionary mechanisms and implications (in the introduction and discussion) and focusing on immunity and relevance for vaccine development. I am not completely convinced by the LD analyses and the statement that the preferred allelic combinations were selected. Could the observed LD not just be caused by occurrence of one given mutation on a specific background? Figure 4 is not informative enough in this regards as it does not provide any information on the variants' AF that will also influence r^2 . A plot of LD decay with distance (or the estimation of the population recombination rate for each gene) might also better emphasize the distinctiveness of CSP (compared to the two other genes).

I got very confused by the simulations and am still not sure I fully understand them. Since the authors seem to estimate COI directly from the amplicon sequencing data, why is the COI different from the number of haplotypes on Figure 5? I think the authors actually test whether, for example, two haplotypes drawn randomly from the pool are more different from each other than two haplotypes observed in an actual infection. Is this correct? This analysis should be better explained in the main text. Also for this analysis and the association with patient age, it would be interesting to compare the results obtained for these candidate genes with COI estimates obtained for neutral markers (if these are available).

The authors should rephrase the interpretation of the last results on the acquisition of immunity in sporozoite vs. blood stage parasites (lines 363-367) that is not supported by the current data. As implied by the authors, it is very possible that other epitopes may yield very different results and the extrapolation of the response against SERA2 vs CSP or TRAP is overgeneralized.

Minor comments

The number of infections analyzed that presented throughout the manuscript is a little misleading. The authors should clarify the sample sizes (that are different for the different loci) and change the number in the abstract (from "over 4,500" to "over 2,800")

I am a bit surprised by the amount of "highly variant sites". Given that some of the genes targeted belong to multigene families, is it possible that some of these variants are caused by collapsing orthologous sequences? Are these variants also detectable in monoclonal infection (i.e., where there is a single haplotype)

On Figure 1: Replace "cell count" by "# of parasites"

In Table 2: remove the percentiles (from the right part of the table) and include the actual percentiles in brackets for each value.

On line 190: change to "overlapping MiSeq read pairs". I initially understood that each locus was sequenced using multiple overlapping amplicons.

David Serre

Reviewer #2 (Remarks to the Author):

This manuscript describes the analysis of data generated by deep sequencing of fragments of

three *Plasmodium falciparum* genes, using >4,000 independent infection samples obtained during a recent Phase III vaccine trial in Africa. The authors use innovative approaches to compare intra-infection diversity with inter-infection diversity in order to identify residues or regions that may be targets of strain-specific immune responses. Strain-specific immunity is a significant issue for *P. falciparum* blood-stage vaccine development in particular, and the approach outlined here is a new and potentially interesting angle to identifying individual residues under strong pressure. The quality of the data, figures and writing are strong, but enthusiasm is significantly diminished by the choice of genes being studied, which do not generate any particularly novel insights. What is left therefore is an intriguing proof of principle, which might yield more if pointed at novel candidates or high priority vaccine targets.

Major points

1. Figure 1 is slightly misleading. From the Methods it appears that >500 genes have been excluded from whole-genome analysis in part because they give high levels of heterozygosity. Genes with high levels of heterozygosity are likely the most variable genes in the genome - doesn't the approach of excluding these genes therefore inflate the apparent diversity of CSP, TRAP and SERA2?
2. SERA2 is a very odd choice for a blood stage antigen. In vitro data suggests that it is not strongly expressed in blood stages at the protein level, although this study was carried out in only a single strain, so caveats are needed - however, there are clearly other members of the SERA family that are much more strongly expressed. While antibodies against it have been detected, it has not, unlike CSP and TRAP, been extensively profiled in immunoepidemiology studies. Finally and most significantly, if the aim of the study (as suggested in the Introduction) was to explore strain-specific immunity, choosing blood stage vaccine candidates where this is known to be an issue (MSP2, AMA1, EBA175), or novel high priority blood stage candidates that have not been extensively studied (Rh5, RIPP) would seem much more interesting. SERA2 is a very obscure choice, and it is difficult to know how much can be extrapolated from this to other candidates. Given that strain-specific responses are most likely a significant problem in the blood stages, and this is the only blood stage candidate studied, the choice significantly diminishes the impact and interest of the manuscript.
3. The question of intra-host vs. inter-host diversity is in many ways the crux of the paper, and is a potentially very interesting approach. In analysing the findings, much is made of two amino acids in CSP, where intra-host frequencies deviate significantly (one positively, one negatively) from those expected based on whole population analyses. Perhaps more surprising is that none of the amino acid's in SERA2 deviate significantly from expected frequencies, and most of those in both CSP and TRAP do not either. This seems to be in contrast to studies of Tajima's D on CSP for example, which highlight much larger regions that are potentially under balancing selection. Why the disparity between this new approach and previous studies? Is allele-specific immunity much less frequent than thought? Or is this an issue with the cut-offs being used in the models developed here?
4. Age effect. The study uses samples from five sites and three countries, which may have very different transmission intensities, meaning that the acquisition of immunity may be quite different between them. Have the authors tried separating the samples based on geographic location and re-running the age analysis to see whether stronger effects emerge in some individual locations than in the entire pooled sample set?

Minor points

1. Figure 1 is all very well as an indicator, but gives no sense of relative expression - is SERA2 abundantly expressed in blood stages, for example? There have been many comparative RNAseq and microarray studies from which this information could be drawn and added to the figure to give a more nuanced sense of the potential immune exposure of each antigen.
2. Mapping variable sites to protein structures. While no structure is available for SERA2 to visualise where variable residues are located, a structure is available for the homologous protein SERA5. It is possible to model the SERA2 amplified region using this SERA5 structure, and map variable residues against the model?

3. The Methods note that reads with “evidence of PCR chimerism” were removed from analysis. This could be a major potential confounder especially for the LD analysis. The section describing how chimerism was identified and eliminated needs expanding.

Response to Reviewers

Early et al. "Within-infection diversity of *Plasmodium falciparum* antigens reflects host-mediated selection."

We thank the reviewers for their attentive reading and suggestions, which we feel have strengthened the manuscript. Below, we detail the changes we have made in this revised version.

Reviewer #1 (Remarks to the Author):

The manuscript by Early et al. describes the analyses of >2,800 *P. falciparum* infections using targeted amplicon sequencing for three antigens. The amount of data generated is very impressive and the analyses interesting (though they could sometimes be explained more simply and convincingly). I have some specific concerns but, overall, a very exciting study.

I do not understand the emphasis on balancing selection. For example, on line 146, diversifying selection should probably be a better suited terminology than balancing selection, which is very specific and may not reflect the situation described by the authors. Also, balancing selection is classically associated with positive Tajimas D values and two of the three genes targeted have clear negative values (though slightly less negative than the genome average). I would recommend replacing this term throughout the manuscript.

We have added or substituted the term "diversifying selection" throughout the text.

In general, I think the paper would be improved by simplifying the discussion on the evolutionary mechanisms and implications (in the introduction and discussion) and focusing on immunity and relevance for vaccine development.

We have reordered the points made in the Discussion to place more emphasis on our results' applicability to vaccine development, and have reworded a few sections in the Introduction with the same goal in mind. However, we did not fully remove any of the discussion of evolutionary mechanisms. As the analyses we present are evolutionary in nature, we believe these discussion points provide necessary context for properly interpreting our results.

I am not completely convinced by the LD analyses and the statement that the preferred allelic combinations were selected. Could the observed LD not just be caused by occurrence of one given mutation on a specific background? Figure 4 is not informative enough in this regards as it does not provide any information on the variants AF that will also influence r^2 . A plot of LD decay with distance (or the estimation of the population recombination rate for each gene) might also better emphasize the distinctiveness of CSP (compared to the two other genes).

Within our dataset, there are rare alleles that are confined to a single haplotypic background, however, we excluded from the LD figure sites with only rare alleles (major allele frequencies > 0.98).

As the reviewer suggested, we have added a plot of LD by distance to the Supplementary Materials (Figure S2). This figure shows that, compared to *TRAP* and *SERA2*, *CSP* contains nucleotide pairs with unusually high LD given the distance between them.

We have also added Supplementary Table S4, which gives the number of unique haplotypes containing each nucleotide pair. These numbers provide more information on these pairs and show which ones are only present on single haplotypes. Note that for pairs of particular interest (e.g. 950-1060), allelic combinations are found on multiple haplotypic backgrounds and were sampled from all study sites. This suggests that in most cases the LD observed between these pairs of sites cannot be trivially explained by the recent origin of one or both variants and insufficient time for recombination to have occurred.

I got very confused by the simulations and am still not sure I fully understand them. Since the authors seem to estimate COI directly from the amplicon sequencing data, why is the COI different from the number of haplotypes on Figure 5? I think the authors actually test whether, for example, two haplotypes drawn randomly from the pool are more different from each other than two haplotypes observed in an actual infection. Is this correct? This analysis should be better explained in the main text.

We have rewritten the text and revised Figure 5 to more clearly describe this analysis. We now explicitly define our measure of intra-host diversity and state how it differs from COI. In the figure and the text, we have replaced “COI” with the phrase “haplotype count” or “haplotype number” to show that we are referring to the number of unique amplicon haplotypes, which could differ slightly from the number of unique parasite lineages (COI).

Also for this analysis and the association with patient age, it would be interesting to compare the results obtained for these candidate genes with COI estimates obtained for neutral markers (if these are available).

We agree this would be an interesting comparison, but unfortunately, we do not have genotyping data for neutral loci. This is one reason why a comparison of the three proteins is interesting, however. The observation that age correlates with intra-host diversity at SERA2 but not TRAP or CSP suggests that the strength of the age relationship varies across different proteins.

The authors should rephrase the interpretation of the last results on the acquisition of immunity in sporozoite vs. blood stage parasites (lines 363-367) that is not supported by the current data. As implied by the authors, it is very possible that other epitopes may yield very different results and the extrapolation of the response against SERA2 vs CSP or TRAP is overgeneralized.

We have removed the comparison of SERA2 to CSP and TRAP that was in this paragraph.

Minor comments:

The number of infections analyzed that presented throughout the manuscript is a little misleading. The authors should clarify the sample sizes (that are different for the different loci) and change the number in the abstract (from over 4,500 to over 2,800)

We removed the number from the abstract and use the lower number in the Introduction. To further clarify the sample sizes in the text, we added a column for “Sampled infection” counts in Table 2.

I am a bit surprised by the amount of highly variant sites. Given that some of the genes targeted belong to multigene families, is it possible that some of these variants are caused by collapsing

orthologous sequences? Are these variants also detectable in monoclonal infection (i.e., where there is a single haplotype)

The sequenced amplicon regions are sufficiently diverged from other *P. falciparum* genes so that inadvertent sequencing of paralogs is not an issue. Using BLAST to look for homologous regions elsewhere in the genome, we found the following maximal coverage of the amplicons: *CSP*--68%, *TRAP*--68%, *SERA2*--15%. Only full-length amplicons were used in the analyses, so even if sequenced, none of these regions of homology would have been retained.

A major reason for the large number of highly variant sites is the deep sampling approach. As shown in Supp Tables S1-S3, many of the individual variants are at very low frequency and would not have been recovered with smaller sample sizes. These sites would have therefore been labeled only bi-allelic with shallower sampling.

We have checked whether haplotypes or alleles are differentially distributed between monoclonal and polyclonal infections. There is no evidence that this has occurred.

On Figure 1: Replace cell count by # of parasites

We have changed this wording as suggested.

In Table 2: remove the percentiles (from the right part of the table) and include the actual percentiles in brackets for each value.

We have made this formatting change.

On line 190: change to overlapping MiSeq read pairs. I initially understood that each locus was sequenced using multiple overlapping amplicons.

This wording has been changed as suggested.

Reviewer #2 (Remarks to the Author):

This manuscript describes the analysis of data generated by deep sequencing of fragments of three *Plasmodium falciparum* genes, using >4,000 independent infection samples obtained during a recent Phase III vaccine trial in Africa. The authors use innovative approaches to compare intra-infection diversity with inter-infection diversity in order to identify residues or regions that may be targets of strain-specific immune responses. Strain-specific immunity is a significant issue for *P. falciparum* blood-stage vaccine development in particular, and the approach outlined here is a new and potentially interesting angle to identifying individual residues under strong pressure. The quality of the data, figures and writing are strong, but enthusiasm is significantly diminished by the choice of genes being studied, which do not generate any particularly novel insights. What is left therefore is an intriguing proof of principle, which might yield more if pointed at novel candidates or high-priority vaccine targets.

Major points

1. Figure 1 is slightly misleading. From the Methods it appears that >500 genes have been excluded from whole-genome analysis in part because they give high levels of heterozygosity. Genes with high levels of heterozygosity are likely the most variable genes in the genome – doesn't the

approach of excluding these genes therefore inflate the apparent diversity of CSP, TRAP and SERA2?

We understand the reason for this concern and revised the wording in the Methods to clarify how and why the gene filtration was done. The genes masked in the initial sample filtration step were later re-inserted into the analysis. In the final dataset, 315 genes were masked due to high levels heterozygosity and/or missing data in the GATK calls. As our analysis was limited to single-clone infections, we expected these samples to have very few true heterozygous calls. Therefore, high sample-level heterozygosity likely resulted from errors in aligning reads to the reference genome assembly.

Overall, this filtering step does not substantially impact the results. If we calculate genome-wide percentiles for nonsynonymous polymorphism without excluding these genes, we still find that *CSP* and *TRAP* are in the upper 99th percentile while *SERA2* is in the 95th. All three amplicon regions remain in the 99th.

2. SERA2 is a very odd choice for a blood stage antigen. In vitro data suggests that it is not strongly expressed in blood stages at the protein level, although this study was carried out in only a single strain, so caveats are needed - however, there are clearly other members of the SERA family that are much more strongly expressed. While antibodies against it have been detected, it has not, unlike CSP and TRAP, been extensively profiled in immunoepidemiology studies. Finally and most significantly, if the aim of the study (as suggested in the Introduction) was to explore strain-specific immunity, choosing blood stage vaccine candidates where this is known to be an issue (MSP2, MSP2, AMA1, EBA175), or novel high priority blood stage candidates that have not been extensively studied (Rh5, RIPR) would seem much more interesting. SERA2 is a very obscure choice, and it is difficult to know how much can be extrapolated from this to other candidates. Given that strain-specific responses are most likely a significant problem in the blood stages, and this is the only blood stage candidate studied, the choice significantly diminishes the impact and interest of the manuscript.

We agree that our analyses of CSP and TRAP may carry more immediate impact than that of SERA2, given the greater investment by previous studies into those loci. As this study takes a new analytical approach, however, it was designed to both provide novel insight and serve as a proof-of-concept. We chose to profile *SERA2* on the basis of population genetic evidence for its immunogenicity. As figure 2A illustrates, the region of SERA2 we profiled with amplicon sequencing exhibits high level of diversity, suggesting it is subject to immune-mediated selection pressure, irrespective of its lower expression level relative to other members of the SERA family. Rh5, while inarguably a more ripe vaccine target, does not exhibit a comparable level of diversity and would therefore not be as amenable to the analyses we present here. We believe this manuscript now lays the ground work for future studies on a wider range of antigens, including those mentioned by the reviewer above.

Additionally, we did not intend to portray SERA2 as a prototypical example of a blood-stage antigen. We have now clarified in the manuscript that the goal of this study was to investigate selection on antigens in general, not to compare liver-stage and blood-stage antigens. As suggested by Reviewer 1, we have removed the language in the Discussion that tentatively discussed the expected differences between these two gene classes.

3. The question of intra-host vs. inter-host diversity is in many ways the crux of the paper, and is a potentially very interesting approach. In analysing the findings, much is made of two amino acids in CSP, where intra-host frequencies deviate significantly (one positively, one negatively) from those expected based on whole population analyses. Perhaps more surprising is that none of the amino acids in SERA2 deviate significantly from expected frequencies, and most of those in both CSP and TRAP do not either. This seems to be in contrast to studies of Tajima's D on CSP for example, which highlight much larger regions that are potentially under balancing selection. Why the disparity between this new approach and previous studies? Is allele-specific immunity much less frequent than thought? Or is this an issue with the cut-offs being used in the models developed here?

These are all interesting questions that largely speak to our initial motivation for performing the study. We found that the intra-host and inter-host diversity analyses are complementary and provide different insights into similar questions. Below and in the manuscript, we outline our thoughts on the sources and consequences of these differences:

- **Temporal effects: Inter-host diversity reflects selection that has occurred across numerous infection cycles. The intra-host analysis is enriched for the effect of a single infection cycle. Differences between the two could exist due to host heterogeneity or evolutionary conflicts that exist between the human and mosquito stages or between human hosts.**
- **Genomic window size: Statistics like Tajima's D must be calculated in windows across groups of SNPs. Our intra-host analysis can be conducted on single SNPs and therefore provides a level of granularity that cannot be attained with Tajima's D .**
- **Significance cutoffs: As the reviewer points out, significance cutoffs certainly impact the number of significant sites we identify in the intra-host analysis. We see these significance thresholds as guidelines for future validation studies, not hard delimiters of biological effect.**

In addition, there are several reasons why the intra-host diversity analysis differs among genes:

- **Epistatic effects: Allele-specific immunity could result from recognition of single amino acids or groups of amino acids. Differences at this recognition level could affect our ability to detect selection at individual amino acids (CSP and TRAP) versus amplicon-wide selection (SERA2).**
- **Heterogeneity among hosts: Our inter-host diversity approach will identify selection at individual amino acid positions if the same position experiences selection within many hosts. If different hosts all recognize different loci, this approach will be too underpowered to yield significant results. This host-level heterogeneity may explain why we did not find significant amino acid positions in SERA2 even though intra-host diversity across the whole SERA2 amplicon decreases with patient age.**

4. Age effect. The study uses samples from five sites and three countries, which may have very different transmission intensities, meaning that the acquisition of immunity may be quite different between them. Have the authors tried separating the samples based on geographic location and re-running the age analysis to see whether stronger effects emerge in some individual locations than in the entire pooled sample set?

We agree that transmission intensity will affect the rate at which adaptive immunity will be acquired in a population. Of the 11 sites studied in the RTS,S/AS01 phase 3 clinical trial, we use the five study sites with the highest transmission intensities. In all five populations

between 53% and 89% of the control children developed clinical malaria during the course of the study. We clarified in the Methods that these are all high-transmission areas.

Anecdotally, the study site with the lowest transmission intensity (Agogo) does not show a negative correlation between intra-host SERA2 diversity and patient age when analyzed separately. However, as other factors could also contribute to population-level heterogeneity, we did not feel this qualitative pattern was robust. Instead, we chose to present our analysis as a composite across populations because we are seeking to describe patterns that are consistent across geographic areas.

Minor points

1. Figure 1 is all very well as an indicator, but gives no sense of relative expression - is SERA2 abundantly expressed in blood stages, for example? There have been many comparative RNAseq and microarray studies from which this information could be drawn and added to the figure to give a more nuanced sense of the potential immune exposure of each antigen.

We have added information on expression level to the Figure 1 legend. Since the “Parasite Number” curve is a rough approximation, we would like the Figure itself to serve as descriptive cartoon of general trends. For this reason, we chose not to add this level of detail to the actual figure.

2. Mapping variable sites to protein structures. While no structure is available for SERA2 to visualise where variable residues are located, a structure is available for the homologous protein SERA5. It is possible to model the SERA2 amplified region using this SERA5 structure, and map variable residues against the model?

The crystal structure has been resolved for the central protease-like domain of SERA5. Unfortunately, our SERA2 amplicon does not align to this region.

3. The Methods note that reads with evidence of PCR chimerism were removed from analysis. This could be a major potential confounder especially for the LD analysis. The section describing how chimerism was identified and eliminated needs expanding.

We have added details regarding our chimera filter to the Methods.

REVIEWERS' COMMENTS:

Reviewer #1 (Remarks to the Author):

The authors have convincingly addressed most of my previous concerns and I only have minor comments that should be easy to fix.

In the simulations used to estimate intra-infection diversity, it is not clear how the authors treat identical haplotypes: identical haplotypes cannot be differentiated in the observed data but may arise in the simulations and bias the findings. The authors' result seem to go in the other direction (lower diversity in the observed than simulated data) but this should be more clearly addressed in the manuscript.

On line 140, it is not obvious to me that immune selection is the only mechanism that could result in elevated polymorphism. The sentence should be modified.

The gene names in Table 1 need to be centered.

Reviewer #2 (Remarks to the Author):

The authors have responded comprehensively and carefully to the comments of both reviewers, and the manuscript is clearer and more balanced as a result. No further concerns.

Response to Reviewers

Early *et al.* Host-mediated selection impacts the diversity of *Plasmodium falciparum* antigens within infections

Below we detail the changes we made in response to the reviewers' comments.

Reviewer #1 (Remarks to the Author):

The authors have convincingly addressed most of my previous concerns and I only have minor comments that should be easy to fix.

In the simulations used to estimate intra-infection diversity, it is not clear how the authors treat identical haplotypes: identical haplotypes cannot be differentiated in the observed data but may arise in the simulations and bias the findings. The authors' result seem to go in the other direction (lower diversity in the observed than simulated data) but this should be more clearly addressed in the manuscript.

To clarify this point, we revised the relevant sentence in the Methods and added a sentence in the Results:

“We could not determine whether natural infections contained multiple parasite lineages with identical haplotypes, so we ensured that the simulated infection did not contain duplicate haplotypes.”

On line 140, it is not obvious to me that immune selection is the only mechanism that could result in elevated polymorphism. The sentence should be modified.

We have changed this sentence to: “Its high polymorphism across sub-Saharan Africa suggests that host immune recognition of this protein may affect parasite fitness.”

The gene names in Table 1 need to be centered.

We have made this edit.

Reviewer #2 (Remarks to the Author):

The authors have responded comprehensively and carefully to the comments of both reviewers, and the manuscript is clearer and more balanced as a result. No further concerns.